 

# Repression by PRDM13 is critical for generating precision in neuronal identity

**Bishakha Mona[1][†], Ana Uruena[1][†], Rahul K Kollipara[2], Zhenzhong Ma[1], Mark D Borromeo[1], Joshua C Chang[1], Jane E Johnson[1,3]***

[1]Department of Neuroscience, UT Southwestern Medical Center, Dallas, United States; [2]McDermott Center for Human Growth and Development, UT Southwestern Medical Center, Dallas, United States; [3]Department of Pharmacology, UT Southwestern Medical Center, Dallas, United States

**Abstract** The mechanisms that activate some genes while silencing others are critical to ensure precision in lineage specification as multipotent progenitors become restricted in cell fate. During neurodevelopment, these mechanisms are required to generate the diversity of neuronal subtypes found in the nervous system. Here we report interactions between basic helix-loop-helix (bHLH) transcriptional activators and the transcriptional repressor PRDM13 that are critical for specifying dorsal spinal cord neurons. PRDM13 inhibits gene expression programs for excitatory neuronal lineages in the dorsal neural tube. Strikingly, PRDM13 also ensures a battery of ventral neural tube specification genes such as *Olig1*, *Olig2* and *Prdm12* are excluded dorsally. PRDM13 does this via recruitment to chromatin by multiple neural bHLH factors to restrict gene expression in specific neuronal lineages. Together these findings highlight the function of PRDM13 in repressing the activity of bHLH transcriptional activators that together are required to achieve precise neuronal specification during mouse development.

DOI: https://doi.org/10.7554/eLife.25787.001

*For correspondence:
Jane.Johnson@UTSouthwestern.edu

[†]These authors contributed equally to this work

**Competing interests:** The authors declare that no competing interests exist.

## Introduction

The process of progenitors undergoing cell fate decisions to determine specific cellular identity and tissue patterning is fundamental in the field of developmental biology. Neural development is an excellent model system to study cell fate processes to understand how neural restricted progenitors differentiate into diverse types of neurons through complex interactions of transcriptional activators and repressors. Multiple regulatory design principles involving combinations of transcription factors have emerged through studies from vertebrate and invertebrate model systems that have significant repercussions in determining and maintaining cell identity (*Deneris and Hobert, 2014*; *Dessaud et al., 2008*; *Lai et al., 2016*; *Matise, 2013*). During progenitor fate specification in the developing nervous system, extensive cross-repressive activities between TFs expressed in neighboring domains in the vertebrate neural tube provide bistable switches needed for generating precise cellular identities (*Briscoe et al., 2000*; *Gowan et al., 2001*; *Muhr et al., 2001*). An additional cell fate design principle was recently demonstrated in the generation of progenitor diversity and in terminal differentiation genes that involves the intersection of a broadly expressed transcriptional activator with progenitor-specific repressor proteins (*Kerk et al., 2017*; *Kutejova et al., 2016*; *Nishi et al., 2015*). In contrast, cell fate specification in the dorsal neural tube may represent a modification of these principles where the activity of progenitor-specific transcriptional activator proteins must be suppressed to limit expression of inappropriate genes. Here we identify the transcriptional repressor, PRDM13, as a critical factor in keeping ventral cell-type specification genes silenced in the dorsal neural tube, highlighting the importance of a balanced network of transcriptional activators

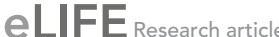 

and repressors for generating precision in neuronal identities throughout populations in the dorsal ventral axis of the neural tube.

The dorsal spinal cord functions as an essential center for integration of somatosensory information (*Liu and Ma, 2011*; *Ross, 2011*). Appropriate processing of this information requires the correct composition of neuronal subtypes, and regulated generation of this neuronal diversity relies on discrete patterns of expression of basic helix-loop-helix (bHLH) and homeodomain (HD) TFs (*Lai et al., 2016*) (*Figure 1A*). In the dorsal neural tube, genetic studies have identified the bHLH factors ASCL1 and PTF1A as promoters of excitatory and inhibitory neuronal fate, respectively (*Glasgow et al., 2005*; *Gowan et al., 2001*; *Helms et al., 2005*; *Mizuguchi et al., 2006*; *Nakada et al., 2004*). The bHLH TFs execute their function by directly promoting transcription of genes encoding HD TFs, as well as other genes for neuronal function (*Borromeo et al., 2014*). ASCL1 induces the excitatory neuron specification HD factors TLX1 and TLX3, while PTF1A induces the inhibitory neuron specification HD factor PAX2 (*Batista and Lewis, 2008*; *Borromeo et al., 2014*; *Chang et al., 2013*; *Cheng et al., 2004*; *Cheng et al., 2005*; *Glasgow et al., 2005*; *Mizuguchi et al., 2006*) (*Figure 1B*). These studies highlight the essential role bHLH TFs play in determining neuronal diversity in the dorsal spinal cord and serve to demonstrate some basic principles of cell fate specification. In order to drive a multipotent progenitor to a specific lineage, the bHLH TFs must be capable of promoting the fate-specific program while simultaneously suppressing the opposing transcriptional programs for alternative fates (*Gowan et al., 2001*; *Helms et al., 2005*). The all-or-nothing nature of this genetic switch is apparent in that gain or loss of function of either factor does not give rise to neurons of mixed identity. Since both ASCL1 and PTF1A function as transcriptional activators (*Beres et al., 2006*; *Nakada et al., 2004*), the mechanism for repression was discovered to be indirect, involving PTF1A increasing levels of the transcriptional repressor PRDM13 (*Chang et al., 2013*). Thus, a network of TFs including bHLH and PRDM factors control levels of HD factors that generate the necessary balance of excitatory and inhibitory neurons in the dorsal spinal cord that modulate somatosensory information processing (*Figure 1A,B*).

PRDM13 is a transcriptional repressor present in progenitor domains of the dorsal but not the ventral neural tube from hindbrain regions to the spinal cord (*Chang et al., 2013*) (*Figure 1C*). It is a member of the PRDM family of proteins defined by an N-terminal PR domain and a varying number of C-terminal zinc-finger (ZF) domains (*Hohenauer and Moore, 2012*). PRDM factors function in a variety of cell lineages to promote cell proliferation and cell type specification (*Bikoff et al., 2009*; *Eom et al., 2009*; *Huang et al., 1998*; *Ross et al., 2012*). Mechanistically they rely on protein-protein and protein-DNA interactions through the PR and ZF domains to repress or activate transcription. The role of PRDM13 as a cell type specification factor has been demonstrated in neural tube (*Chang et al., 2013*) and retina (*Watanabe et al., 2015*). In retina, PRDM13 functions to specify subsets of retinal amacrine interneurons. Within the dorsal spinal cord, PRDM13 functions as a downstream effector of PTF1A to suppress the excitatory neuronal program within the inhibitory neuronal lineage. Several PRDM family members have been shown to have intrinsic histone methyltransferase activity mediated by the PR domain (*Di Zazzo et al., 2013*; *Fog et al., 2012*; *Fumasoni et al., 2007*; *Hohenauer and Moore, 2012*). PRDM13 has been suggested to have intrinsic histone methyltransferase activity, although the relevance of this in vivo remains unclear (*Hanotel et al., 2014*) since the overexpressed PRDM13 ZF domain is sufficient to specify inhibitory neurons in retina and neural tube (*Chang et al., 2013*; *Watanabe et al., 2015*).

To further probe the function of PRDM13 in spinal cord development, and to uncover molecular mechanisms that determine neuronal diversity, we generated multiple *Prdm13* mutant mouse strains. In contrast to the previously reported *Prdm13* mutant deleting exons 2 and 3 (*Watanabe et al., 2015*), *Prdm13* mutant mice generated here are neonatal lethal and appear to reflect a null phenotype. Analysis of the neural tube phenotype confirms the role of PRDM13 in regulation of neuronal specification factors PAX2 for inhibitory neuronal lineages and TLX1/3 for excitatory neuronal lineages during early neurogenesis (*Chang et al., 2013*). However, unbiased approaches probing gene expression changes in mutant neural tubes reveal much broader roles for PRDM13 in neuronal specification. These include a negative feedback loop between PRDM13 and its activator, PTF1A, and a requirement for PRDM13 in direct suppression of multiple ventral neuronal-subtype specification transcription factors. Mechanistically, PRDM13 is bound to genomic regions overlapping those bound by neural bHLH factors, and PRDM13 represses the ability of these bHLH factors to activate enhancer driven reporters. Together, a model is presented for recruitment of PRDM13 to bHLH



**Figure 1.** Generation of multiple mutant mouse alleles of *Prdm13*. (**A**) Diagram of half an E10.5 neural tube highlighting expression domains of TFs used in this study. The ventricular zone containing neural progenitors (left) and differentiated neuronal populations (right) are shown. See (*Lai et al., 2016*) for a more comprehensive picture of the TFs patterning the neural tube at this stage. (**B**) Transcription factor network functioning in the dorsal neural tube generating inhibitory and excitatory neurons. (**C**) *Prdm13* mRNA is restricted to the dorsal neural tube in mice at E11.5. (**D–E**) Three different targeting strategies resulting in *Prdm13^GFP^*, *Prdm13^ΔZF^*, and *Prdm13 ^Δ115^* mutant mouse strains. PCR on tail DNA with primers P1-P5 confirm the genotype of the mutant mice. Red box indicates an inserted stop codon. Yellow lines indicate individual ZFs. (**F–I**) Immunofluorescence showing PRDM13 (magenta) in wildtype (WT) and *Prdm13* mutant strains in E11.5 neural tube. (**F'–I'**) Higher magnification including DAPI (green) to label nuclei. (**J**) Western blot from verifying loss of PRDM13 protein in neural tube of *Prdm13^GFP/GFP^* mutants and the expression of the mutant PRDM13 protein in *Prdm13 ^Δ115/Δ115^* mutants. PRDM13 detected in wildtype mouse neural tube and telencephalon are positive and negative controls, respectively. βactin is the loading control.

DOI: https://doi.org/10.7554/eLife.25787.002

bound enhancers to ensure silencing of lineage inappropriate factors, a function that promotes precision in generating neuronal subtype specific fates.

## Results

To understand the role PRDM13 plays in spinal cord development in vivo, and to uncover mechanistic insights into PRDM13 activity, we generated multiple mutant alleles of *Prdm13* in mice. These include a GFP knockin (*Prdm13^GFP*) that disrupts generation of PRDM13, and a deletion within exon 4, the genomic region that encodes the terminal three zinc fingers (ZFs) of PRDM13 (*Prdm13^ΔZF*) (*Figure 1D*). This ZF domain is predicted to be essential for PRDM13 function because in overexpression assays this domain is sufficient to mimic the activity of the full-length protein, and overexpression of mutant PRDM13 variants that disrupt each ZF lose activity relative to wild type (*Chang et al., 2013*). The phenotypes observed in both *Prdm13^GFP* and *Prdm13^ΔZF* mutants are identical. Heterozygous animals are indistinguishable from wild-type littermates, but when homozygous for the mutant allele they die at birth, similar to the neonatal lethality seen in other mutants null for TFs that regulate neuronal differentiation and specification such as the bHLH factors ASCL1 and PTF1A (*Glasgow et al., 2005*; *Guillemot and Joyner, 1993*). PRDM13-specific antibodies generated to the C-terminal domain (*Watanabe et al., 2015*) do not detect protein in these mutants at E11.5 by Western blot or immunofluorescence (*Figure 1F–H,J*). A third mutant was generated with a 115 bp deletion near the N-terminus that causes a frame shift (*Prdm13^Δ115*) (*Figure 1E*). Surprisingly, *Prdm13^Δ115/Δ115* survives into adulthood, similar to a previously reported *Prdm13* mutant that deleted exons 2 and 3 that encodes much of the PR domain of the protein (*Watanabe et al., 2015*). Western analysis and immunohistochemistry reveal that a protein containing at least the C-terminal ZF domain is translated in *Prdm13^Δ115* (*Figure 1I,J*). This mutant PRDM13 has sufficient function for normal survival (see Discussion). To uncover the requirement for PRDM13 in spinal cord development, we focused the remaining experiments on the *Prdm13^GFP* and *Prdm13^ΔZF* mutants.

Dorsal neuronal populations are defined largely by the combination of HD factors they express, and in E10.5 neural tube they are designated as dorsal interneurons 1–6 (dI1-dI6) (for review see [*Lai et al., 2016*]). *Prdm13* was identified as an important specification gene in the dorsal neural tube required for the balance of inhibitory (PAX2^+;LHX1/5^+, dI4) interneurons and excitatory (TLX1/3^+, dI5) interneurons in the chick neural tube (*Chang et al., 2013*). Overexpression of PRDM13 by electroporation into the chick neural tube led to a suppression of TLX1/3 and a subsequent increase in the number of PAX2^+ cells. The latter is thought to be an indirect consequence of TLX1/3 function in repressing PAX2 (*Chang et al., 2013*; *Cheng et al., 2004*; *Cheng et al., 2005*). Support for these functions of PRDM13 is seen in *Prdm13^GFP* and *Prdm13^ΔZF* mutant embryos with the loss of PAX2^+; LHX1/5^+ dI4 neurons and an increase in TLX1/3^+ dI5 neurons in the dorsal neural tube. At E10.5, antibodies specific for these factors were used to examine changes in the dorsal interneuron populations (*Figure 2*). A complete loss of dI4 (PAX2^+;LHX1/5^+) and an increase in dI3/5 (TLX1/3^+) in both the *Prdm13^GFP/GFP* and *Prdm13^ΔZF/ΔZF* neural tubes was seen at this stage. These phenotypes are identical to the *Ptf1a* null mouse (*Figure 2G*) (*Glasgow et al., 2005*), and are opposite to the *Prdm13* overexpression phenotypes in the chick (*Chang et al., 2013*). Thus, these findings confirm PRDM13 functions as a repressor of the excitatory neuronal gene program within the inhibitory neuronal lineage in the dorsal neural tube.

The *Prdm13* mutant phenotype, however, does not completely mimic that seen for *Ptf1a* mutants in the dorsal neural tube. One important distinction is the marked reduction of the dI2 population (LHX1/5^+; PAX2^-) in the *Prdm13* mutants that is not seen with the loss of *Ptf1a* (*Figure 2*). This is not unexpected given that there is *Ptf1a*-independent expression of *Prdm13* that places it in a broader domain than that of *Ptf1a* (*Chang et al., 2013*) (*Figure 1A*). Another notable difference between the *Prdm13* and *Ptf1a* mutants is detected at later stages of neural tube development. The *Ptf1a* mutants are striking in that there is essentially a complete loss of PAX2^+ neurons in the dorsal spinal cord with a subsequent absence of GABAergic and glycinergic neuronal lineages (*Glasgow et al., 2005*) (*Figure 3*). Both *Prdm13^GFP/GFP* and *Prdm13^ΔZF/ΔZF* neural tubes show only partial loss of PAX2^+ cells dorsally at E12.5 and E16.5, markedly different from the absolute loss of PAX2 observed in the *Ptf1a^CRE/CRE* null (*Figure 3A–G*), and at earlier stages of neurogenesis (*Figure 2*). This partial loss of PAX2^+ cells was accompanied by a modest increase in the number of TLX1/3^+ cells in the *Prdm13* mutants, which was not as dramatic as the increase seen in the *Ptf1a* null (*Figure 3H*).

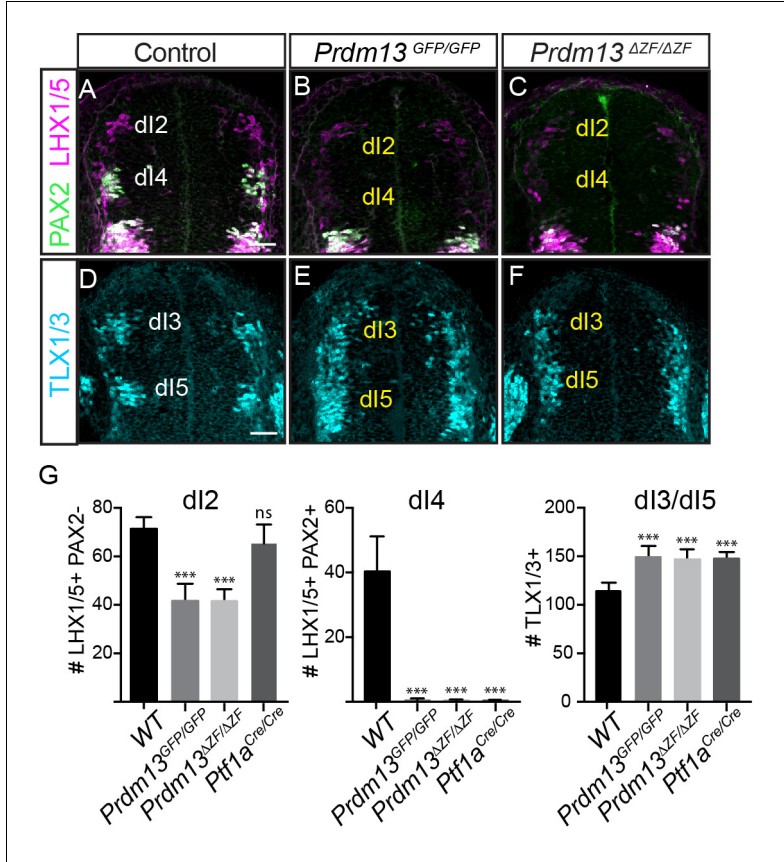

**Figure 2.** PRDM13 is required for specification of dorsal interneurons in the spinal cord. (**A–C**) PAX2 (green) and LHX1/5 (magenta) immunofluorescence show the loss of dI4 and decrease of dI2 interneurons in the *Prdm13* mutants relative to WT at E10.5. (**D–F**) TLX1/3 (cyan) immunofluorescence shows an increase in the number of dI3/5 interneurons in the *Prdm13* mutants relative to WT. Scale bar: 50 µM. (**G**) Quantification reporting the number of dI2 (LHX1/5+;PAX2-), dI4 (PAX2+;LHX1/5+) and dI3/dI5 (TLX1/3+) cells per neural tube section in WT, *Prdm13* mutants, and for comparison, *Ptf1a^Cre* nulls. Error bars indicate SEM. Student's t-test was used to determine significant differences relative to WT (***p<0.001, NS = not significant) (n ≥ 5 embryos for each genotype).
DOI: https://doi.org/10.7554/eLife.25787.003
The following source data is available for figure 2:

**Source data 1.** PRDM13 is required for specification of dorsal interneurons in the spinal cord.
DOI: https://doi.org/10.7554/eLife.25787.004

Consistent with only a partial loss of PAX2+ cells in the *Prdm13* mutants, *Gad1*, required for synthesis of the inhibitory neurotransmitter GABA, is also only partially lost dorsally as seen by in situ hybridization at E16.5 (**Figure 3I–K**). Again this contrasts to the complete loss of this marker in the *Ptf1a* null animals at these later stages (**Glasgow et al., 2005**). The partial loss of *Gad1* dorsally in the absence of PRDM13 presumably reflects loss of the inhibitory neurons derived from the E10.5 dI4 PAX2[+] population (**Figure 2**). Similarly, only a modest increase in *Vglut2 (Slc17a6)*, a marker of excitatory neurons, is detected in the lateral region of the domain lacking *Gad1* (**Figure 3L–N**). These results highlight overlapping but distinct consequences to dorsal spinal cord development with the loss of PRDM13 versus its upstream regulator PTF1A.

To discover additional PRDM13 functions in neural development, we identified differentially expressed genes (DEGs) in dorsal neural tube cells in the presence and absence of PRDM13. RNA-seq was performed on GFP[+] cells isolated by fluorescence-activated cell sorting from E11.5 *Prdm13^GFP/+* versus *Prdm13^GFP/GFP* neural tubes (**Figure 4A**, **Table 1**, and **Supplementary file 1**). 837 genes had greater than a 1.5-fold difference in expression and an FPKM >1 in at least one of the conditions. Gene ontology (GO) analysis determined that a large number of the DEGs in the

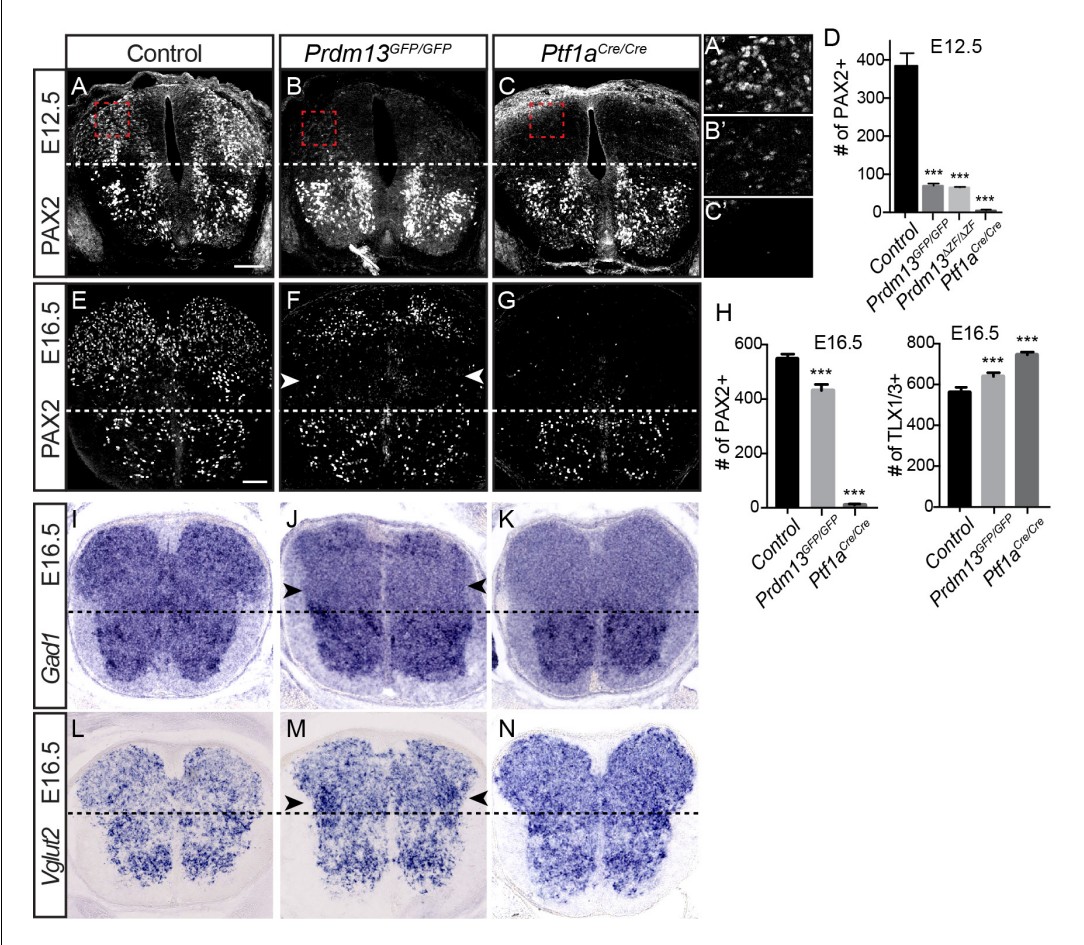

**Figure 3.** PRDM13 is not required for specification of dorsal interneurons during the late wave of neurogenesis. Transverse sections of mouse neural tube. (A–C, E–G) Immunofluorescence for PAX2 highlights the reduced numbers of PAX2 positive neurons in the dorsal neural tube in *Prdm13* mutants compared to no PAX2+ neurons in *Ptf1a* mutants at E12.5 (A–C) and E16.5 (E–G). (A'–C') Higher magnification images of the red box highlighted (A–C). (D,H) Quantification reporting the number of PAX2+ or TLX1/3+ cells per neural tube section in WT, *Prdm13* mutants, and for comparison, *Ptf1a*Cre nulls. Error bars indicate SEM. Student's t-test was used to determine significant differences relative to WT (***p<0.001) (n ≥ 8 for each genotype). Scale bar: 50 µM. (I–N) In situ hybridizations for *Gad1* mRNA (I–K) shows the loss of inhibitory neurons in *Prdm13* and *Ptf1a* mutants which is compensated by increase in excitatory neurons shown by *Vglut2 (Slc17a6)* mRNA (L-N arrowheads) at E16.5. Note the loss of dorsal *Gad1* is more complete in the *Ptf1a* versus the *Prdm13* mutants (J,K).

DOI: https://doi.org/10.7554/eLife.25787.005

The following source data is available for figure 3:

**Source data 1.** PRDM13 is not required for specification of dorsal interneurons during the late wave of neurogenesis.
DOI: https://doi.org/10.7554/eLife.25787.006

*Prdm13* mutants play a role in cell proliferation, neuronal specification and cell cycle regulation (**Supplementary file 2**).

We also performed PRDM13 ChIP-Seq to identify candidate direct gene targets of PRDM13 to advance our mechanistic understanding of its function. To increase confidence in PRDM13 bound sites identified, we performed ChIP-seq experiments using three different antibodies (see Materials and methods), and used only those sites identified in at least two experiments. We identified 220 PRDM13 binding sites largely located at distal sites relative to the transcription start sites of genes. These targets included a known target of PRDM13, *Tlx3* (**Chang et al., 2013**). The genome tracks for the *Tlx3* locus showing RNA-Seq and ChIP-Seq data illustrate the increase in *Tlx3* transcripts in the dorsal neural tube of *Prdm13* mutants relative to wild type, and the binding of PRDM13 to a region >20 kb 5' of *Tlx3* (**Figure 4B**). This region was previously shown to drive expression of a



**Figure 4.** PRDM13 binds to the genome at many sites also bound by ASCL1 and PTF1A. (**A**) Strategy to determine transcriptional targets of PRDM13 in E11.5 neural tubes. Venn diagrams illustrate the number of PRDM13 bound sites from ChIP-Seq (220), the genes associated with these sites (384), the genes upregulated in neural tubes from *Prdm13*<sup>GFP/GFP</sup> versus *Prdm13*<sup>GFP/+</sup> (455), and the genes downregulated in mutant neural tubes (356). The overlap represents PRDM13 targets (60) increased (54, p-value=1.26e-27) and decreased (6, p-value=0.16) in *Prdm13* mutants relative to controls. (**B**) Genome tracks at the *Tlx3* locus showing the RNA-Seq from *Prdm13* mutants, the PRDM13 ChIP-Seq, and mammalian sequence conservation. The known enhancer of *Tlx3* (*eTlx3*) is highlighted. (**C**) De novo motif enrichment in PRDM13 bound regions (150 bp around each peak summit) using HOMER package v4.2. The motif with the most enrichment is an Ebox found in 40% of the sites. (**D**) Heat map of the 220 PRDM13 (red) bound sites from ChIP-seq from E11.5 mouse neural tube. A subset of these regions is also bound by ASCL1 (blue) and/or PTF1A (green).

DOI: https://doi.org/10.7554/eLife.25787.007

reporter gene to the dI3/dI5 domains in the chick neural tube, identifying it as a *Tlx3* enhancer (*Chang et al., 2013*).

De novo motif analysis failed to identify a novel binding motif unique to PRDM13, but PRDM13 bound sites were enriched for motifs for known neural TFs such as E-box (bHLH) (40%), Rfx (17%), Pdx (HD) (25%) and Sox (30%) motifs (*Figure 4C*). Indeed, almost half of the PRDM13 bound sites overlap with the E-box binding factors ASCL1 (15), PTF1A (48), or both (39) (*Figure 4D*). The lack of a unique binding motif is consistent with a model where PRDM13 is recruited to sites by other site-specific TFs.

PRDM13 peak-to-gene associations were identified using GREAT (*McLean et al., 2010*), which identified 384 genes. GO analysis of these genes also showed enrichment for those that play a role in cell proliferation, neuronal specification and cell cycle regulation (*Supplementary file 2*). 60 putative direct targets of PRDM13 were identified by intersecting these genes with the DEGs identified in *Prdm13*<sup>GFP</sup> neural tubes (*Figure 4A*, *Supplemental file 3*). Consistent with the reported function of PRDM13 as a transcriptional repressor (*Chang et al., 2013*), mRNA levels for 54 of the putative

**Table 1.** Select differentially expressed genes (DEG) in the dorsal neural tube of mouse *Prdm13* mutants (*Prdm13^GFP/+* versus *Prdm13^GFP/GFP*).

Genes with associated PRDM13 ChIP-seq peaks in two of three experiments are bolded. See *Supplementary files 1* and *3* for a full list of DEG and PRDM13 binding site coordinates.

| TFs | Fold up in mutants | Neural related Factors | Fold up in mutants | TFs | Fold down in mutants | Neural related Factors | Fold down in mutants |
|---|---|---|---|---|---|---|---|
| *Olig2* | 1114 | *Calb2* | 7.3 | *Hmx2* | 6.8 | *Gad2* | 6.2 |
| *Olig1* | 217 | *Syt13* | 5.2 | *Pax2* | 6.3 | *Slc6a5* | 5.9 |
| *Prdm12* | 24 | *Slc1a3* | 4.4 | *Lhx1* | 5.7 | *Slc32a1* | 5.6 |
| *Tlx3* | 17 | *Frzb* | 4.2 | *Pax8* | 5.6 | *Nrxn3* | 5.4 |
| *Phox2b* | 17 | *Mfng* | 4.2 | *Hmx3* | 5.5 | *Cacna2d3* | 4.9 |
| *Otp* | 13 | *Slc17a6* | 4.1 | *Lhx5* | 4.4 | *Gad1* | 4.6 |
| *Phox2a* | 13 | *Ddc* | 3.9 | *Gsx2* | 3.4 | *Slc30a3* | 4.4 |
| *Shox2* | 12 | *Slc17a7* | 3.7 | *Skor2* | 3.3 | *Epha2* | 3.4 |
| *Lmx1b* | 12 | *Cbln2* | 3.5 | *Gbx2* | 2.9 | *Gria1* | 3.4 |
| *En1* | 5.1 | *Slc31a2* | 3.6 | *Foxd3* | 2.7 | *Robo4* | 2.9 |
| *Dlx3* | 5.1 | *Ntrk1* | 3.5 | *Mecom* | 2.6 | *Ret* | 2.9 |
| *Dbx1* | 4.8 | *Cbln1* | 3.4 | *Pax5* | 2.6 | *Robo3* | 2.7 |
| *Isl1* | 4.5 | *Slc15a2* | 3.2 | *Gsx1* | 2.3 | *Npy* | 2.6 |
| *Tlx1* | 4.3 | *Sncg* | 3.1 | *Nkx1.1* | 2.1 | *Slc7a5* | 2.4 |
| *Neurod4* | 3.8 | *Chrna4* | 3.1 | *Irx4* | 2.0 | *Slc7a1* | 2.3 |
| *Ptf1a* | 3.5 | *Ncald* | 3.0 | | | *Sez6* | 2.0 |
| *Neurog1* | 3.4 | *Lgr5* | 3.0 | | | | |
| *Pou4f1* | 3.0 | *Cacna2d1* | 3.0 | | | | |
| *Pou3f1* | 2.9 | *Kcnk2* | 2.8 | | | | |
| *Bhlhe22* | 2.8 | *Nrn1* | 2.8 | | | | |
| *Pou4f2* | 2.6 | *Chrna3* | 2.7 | | | | |
| *Prdm13* | 2.6 | *Nphs1* | 2.6 | | | | |
| *Id1* | 2.3 | *Chrnb4* | 2.5 | | | | |
| *Id4* | 2.0 | *Kirrel2* | 2.4 | | | | |
| *Neurog2* | 1.7 | *Nefm* | 2.3 | | | | |

DOI: https://doi.org/10.7554/eLife.25787.008

targets are increased in *Prdm13* mutants relative to controls (p-value=1.26e-27) whereas only six are decreased (p-value=0.16) (*Figure 4A*).

The genes controlled by PRDM13 are notable in multiple respects. First, the function for PRDM13 in repressing the gene expression program for the excitatory neuronal lineage dI5 as was shown in the chick neural tube (*Chang et al., 2013*) is evident in the *Prdm13* mutants. For example, PRDM13 binds the *Tlx3* enhancer and restricts its expression at E10.5/E11.5 (*Figure 4B*, eTlx3). Consistently, *Tlx3* is elevated 17-fold, while *Pax2* is reduced 6.3-fold in the absence of *Prdm13* (*Table 1*). However, RNA-seq in the mutants also uncovered new functions for PRDM13, revealing a much broader function in restricting essential ventral neuronal subtype specification genes from being expressed in the dorsal neural tube (*Table 1*). The following experiments support these conclusions and probe mechanisms of PRDM13 activity in spinal cord development.

## PRDM13 directly represses its upstream regulator *Ptf1a* through the *Ptf1a* auto-regulatory enhancer

*Ptf1a* was upregulated 3.5-fold by RNA-seq in the *Prdm13* mutants relative to control (*Table 1*). To determine the pattern of this increase in expression we performed mRNA in situ hybridization and

immunofluorescence for PTF1A. The increase in PTF1A is localized within its normal dorsal progenitor domain 4 (dP4) (*Figure 5A–F*). Because PTF1A is a transcriptional activator of *Prdm13* transcription (*Chang et al., 2013*) these data reveal a negative feedback loop between PRDM13 and PTF1A (*Figure 5G*) where the product of the activated gene (PRDM13) feeds back to repress transcription of its activator (*Ptf1a*) (*Alon, 2007*).

*Ptf1a* transcription is regulated through a 2.3 kb autoregulatory enhancer (*e2.3Ptf1a*) (*Masui et al., 2008*; *Meredith et al., 2009*) (*Figure 5H*). PTF1A binds and activates transcription through this enhancer in a trimeric transcription activator complex called PTF1 that comprises PTF1A, an E-protein, and RBPJ. Perturbations in the PTF1 trimer binding motifs (E-box plus a TC-box) cause a loss of enhancer activity in reporter assays in chick and in transgenic mice (*Masui et al., 2008*; *Meredith et al., 2009*). One possible mechanism for the negative feedback by PRDM13 is recruitment of PRDM13 to this *Ptf1a* autoregulatory enhancer resulting in blocking activity through this site. Indeed, ChIP-seq experiments demonstrate PRDM13 is bound to this enhancer region (*Figure 5H* and *Figure 5—figure supplement 1*).

We next tested the ability of PRDM13 to regulate transcription through the *Ptf1a* autoregulatory enhancer using the chick electroporation assay and a GFP reporter under the control of the 2.3 kb *Ptf1a* autoregulatory enhancer (*e2.3kb-Ptf1a::GFP*) (*Meredith et al., 2009*). As was previously reported, this enhancer directs GFP expression to the *Ptf1a* domain in the dorsal neural tube and co-electroporation with a *Ptf1a* expression vector dramatically increases enhancer activity (*Figure 5I–J,M*). In contrast, co-electroporation with a *Prdm13* expression vector alone or along with that for *Ptf1a* blocks reporter activity completely (*Figure 5K–M*). Finally, we show that PRDM13 and PTF1A interact, possibly forming a repressor complex, using co-IP assays with lysates from HEK293 cells transfected with FLAG-PRDM13 and MYC-PTF1A (*Figure 5N*). Because PRDM13 also interacts by co-IP with ASCL1 (*Chang et al., 2013*) (*Figure 5O*), we tested whether these interactions are mediated through an E-protein, the common heterodimeric partner for these bHLH factors. We detected no interaction between PRDM13 and the E-proteins E47 and E12 (2 isoforms of *Tcf3*), and HEB (*Tcf12*) (*Figure 5O*). We propose a model whereby PRDM13 directly inhibits *Ptf1a* transcription by interrupting PTF1A auto-regulation through binding with PTF1A to the *Ptf1a* autoregulatory enhancer (*Figure 5P*).

The increased PTF1A expression may be responsible for a difference detected between the *Prdm13* and *Ptf1a* loss of function phenotypes in the second wave of neurogenesis in the dorsal neural tube (*Figure 3*).

## PRDM13 excludes ventral neural tube specification factors from the dorsal neural tube

Examination of the cohort of DEGs from the RNA-Seq analysis predictably revealed a number of transcription factors known to play a role in dorsal neural tube specification (i.e. PAX2, TLX1, TLX3, LHX1, LHX5, LMX1B, ISL1, POU4F1), as well as a number of neural related factors misregulated upon loss of PRDM13 (*Table 1*). Notably, several ventral transcription factors were ectopically expressed within the dorsal neural tube of the *Prdm13* mutant (i.e. OLIG2, PRDM12, PHOX2A, PHOX2B, BHLHB5 (*Bhlhe22*)). The latter finding is striking because these ventral specification factors are restricted to the ventral neural tube. Three factors with the highest fold increase in the *Prdm13* mutant are *Olig1, Olig2* and *Prdm12* (*Table 1*). We used these PRDM13 targets to probe the mechanisms by which PRDM13 represses ventral TFs.

## PRDM13 blocks PTF1A activation of *Olig2* in the dorsal neural tube

OLIG2 is restricted to the motor neuron progenitor domain (pMN) in the ventral spinal cord at these early stages of neurogenesis, and it is required for specification of motor neurons (*Mizuguchi et al., 2001*; *Novitch et al., 2001*; *Zhou et al., 2000*) (*Figure 1A*). Upon loss of PRDM13, the linked genes *Olig1* and *Olig2* show striking ectopic dorsal expression within $Prdm13^{GFP/GFP}$ neural tubes (*Table 1*, 217-fold and 1114-fold, respectively). To determine the pattern of this increase in expression we performed immunofluorescence for OLIG2. The ectopic OLIG2 is localized just lateral to the progenitor domain in the dorsal neural tube (*Figure 6A–C*). This aberrant expression of OLIG2 is not detected in the *Ptf1a* null even though there is a partial loss of PRDM13 in these mutants (*Figure 6D*). Co-labeling of OLIG2 with TLX1/3, PTF1A, and OLIG3 demonstrate the ectopic OLIG2 is restricted to

**Figure 5.** PRDM13 represses transcription of its upstream regulator *Ptf1a* through the *Ptf1a* auto-regulatory enhancer. (A–B) In situ hybridization shows increase in *Ptf1a* mRNA in the *Prdm13* mutant at E10.5 relative to wildtype. (C–F) Immunofluorescence confirms the increased levels of PTF1A in *Prdm13* mutants at E10.5 (D) and E12.5 (F). (G) The transcription network highlighting PRDM13 inhibition of its upstream regulator *Ptf1a*. (H) Genome tracks showing ChIP-Seq data for PRDM13 and PTF1A at the *Ptf1a* locus. The two PTF1A bound sites in the *Ptf1a* autoregulatory enhancer are

*Figure 5 continued on next page*

*Figure 5 continued*

highlighted. The sequence at the center of each PTF1A ChIP peak is shown with the Ebox and TCbox sites boxed, which make up the PTF1 trimer binding motif (shaded). (I–L) Representative images from transverse sections of chick neural tubes co-electroporated with a GFP reporter driven by the *2.3kb-Ptf1a* autoregulatory enhancer with the indicated expression vectors at HH13-14 and harvested 24 hr later. (I'–L') ASCL1 (magenta) is shown as reference for the dP3-dP5 domains. Antibodies to MYC-epitope tag for control or PTF1A, or to PRDM13 are used to show the electroporation efficiency and/or the ectopic expression of PTF1A or PRDM13 (insets). PTF1A induces reporter expression while PRDM13 represses both the endogenous enhancer activity as well as the induced expression driven by PTF1A. (M) Quantification of GFP intensity for experiments shown in (I–L). Student t-test was used for significance relative to control except where indicated, *p<0.05, ***p<0.001. Error bars indicate SEM (n ≥ 8 embryos). (N–O) Western blots using antibodies against MYC-tag or FLAG-tag showing FLAG-immunoprecipitates from HEK293 cells ectopically expressing indicated factors. FLAG-tagged PRDM13 pulls down MYC-tagged PTF1A but not the negative control HOOK3 or the E-proteins (E47, E12, HEB). ASCL1 and HOOK3 are positive and negative controls respectively. (P) Proposed model for PRDM13 inhibition of *Ptf1a* expression by abrogating *Ptf1a* autoregulation. PTF1A functions in a trimeric complex (PTF1) with an E-protein and RBPJ (*Meredith et al., 2009*). Specifics on how PRDM13 interacts with DNA and the complex are not known. Scale bar: 50 μM.
DOI: https://doi.org/10.7554/eLife.25787.009

The following source data and figure supplement are available for figure 5:

**Source data 1.** PRDM13 represses transcription of its upstream regulator *Ptf1a* through the *Ptf1a* auto-regulatory enhancer.
DOI: https://doi.org/10.7554/eLife.25787.011
**Figure supplement 1.** Genome tracks showing ChIP-Seq data for PTF1A and the 3 PRDM13 experiments at the *Ptf1a* locus.
DOI: https://doi.org/10.7554/eLife.25787.010

the dI4 domain (*Figure 6E–H* and *Figure 6—figure supplement 1*). Notably, the ectopic OLIG2 expressing cells co-express TLX1/3 and PTF1A (*Figure 6E–H*), factors that are never co-expressed during normal development, thus indicating a severe disruption of neuronal subtype identity in the dorsal neural tube.

We showed that PRDM13 localizes to a PTF1A bound enhancer and represses enhancer activity (*Figure 5*). To explore whether a similar mechanism is involved in regulating the linked genes *Olig1* and *Olig2*, we identified three genomic regions surrounding the *Olig* genes bound by PRDM13 from ChIP-seq (*Figure 6I* and *Figure 6—figure supplement 1*). All three of these sites, designated as *e1Olig*, *e2Olig* and *e3Olig*, are also bound by PTF1A and/or ASCL1. Of these three sites, only *e2Olig* contains the compound E-box/TC-box motifs that represent the site bound by the PTF1 trimeric complex (*Figure 6J*). We utilized the chick electroporation system to test the ability of each region to drive expression of a GFP reporter gene. All three regions directed expression of GFP in the neural tube, activity that was suppressed by PRDM13 (*Figure 6K–I,O,U,V* and *Figure 6—figure supplement 1*). We tested whether PTF1A or ASCL1 were sufficient to activate the *Olig* enhancers by co-electroporating them individually with each enhancer-reporter. *e2Olig::GFP* but not *e1Olig::GFP* or *e3Olig::GFP* was activated by PTF1A, and this activity was suppressed by PRDM13 (*Figure 6M–O* and *Figure 6—figure supplement 1E,J*). ASCL1 did not increase activity through these enhancers. Although these enhancers do not direct expression restricted to the endogenous *Olig2* domain, reporters require E-boxes within the enhancers for activity (*Figure 6P–T* and *Figure 6—figure supplement 1*). Indeed, *e2Olig::GFP* enhancer activity and its induction by PTF1A is lost when E-boxes that conform to a PTF1 motif are mutated (*Figure 6P,R*).

Our data suggest a model for the requirement for PRDM13 to restrict *Olig1* and *Olig2* expression to the ventral neural tube (*Figure 6W*). Transcriptional activators such as PTF1A can bind and activate transcription through an enhancer located between *Olig1* and *Olig2*. However, the *Olig* genes are not expressed in the PTF1A domain in the dorsal neural tube because PRDM13 is also recruited to these sites to silence transcription. In this way, the *Olig* genes are restricted to the ventral neural tube.

## PRDM13 represses *Prdm12* in the dorsal neural tube via inhibition of NEUROG1 and NEUROG2

*Prdm12* is normally restricted to ventral progenitor domain 1 (p1) and is required for generation of V1 interneurons (*Thélie et al., 2015*) (*Figure 1A*). Like *Olig1* and *Olig2*, *Prdm12* mRNA is strikingly upregulated in the *Prdm13^{GFP/GFP}* mutants (*Table 1*, 24-fold). We verified the ectopic *Prdm12* expression in *Prdm13^{GFP/GFP}* and *Prdm13^{ΔZF/ΔZF}* neural tubes by mRNA in situ hybridization (*Figure 7A–C*). This misregulation of *Prdm12* is not detected in *Ptf1a* mutants (*Figure 7D*). Similar to

**Figure 6.** PRDM13 is required to block *Olig2* expression in the dorsal neural tube via inhibition of PTF1A. (**A–D**) Immunofluorescence shows ectopic OLIG2 in the dorsal neural tube in *Prdm13* but not *Ptf1a* mutants at E11.5. (**E–H**) In the *Prdm13* mutant, the PTF1A[+] and TLX1/3[+] cells are co-expressed with the ectopic OLIG2. (**E′–H′**) High magnification of (**E–H**) highlight the co-expression (white). (**I**) Genome tracks showing ChIP-Seq data from mouse neural tube tissue for PRDM13, PTF1A, and ASCL1 at the *Olig2/Olig1* locus. The PRDM13 peaks are highlighted and a dot above the peak indicates if

*Figure 6 continued on next page*

Figure 6 continued

the site was called as significant (see Materials and methods). Regions *e1Olig-e3Olig* were tested in reporter assays. (J) The sequence at the center of the PRDM13 ChIP peaks contains a compound Ebox/TCbox with the proper spacing, which makes the PTF1 motif, is only present in *e2Olig* (shaded). (K–N, P–S) Representative images from transverse sections of chick neural tubes co-electroporated with *e2Olig::GFP* or *e2Olig-PTF1Mut::GFP* or *e2Olig-allEboxMut::GFP* reporter constructs and expression vectors as indicated at HH13-14 and harvested 24 hr later. Antibodies to MYC-epitope tag or PRDM13 (white) confirm the electroporation efficiency and the overexpression of the respective transcription factor. (K–O) PRDM13 overexpression inhibits *e2Olig::GFP* expression. PTF1A is sufficient to induce *e2Olig2::GFP*, but this activity is repressed by PRDM13. (P–T) *e2Olig::GFP* enhancer activity and its induction by PTF1A is lost when E-boxes that conform to the PTF1 motif or all Eboxes are mutated. (O,T) Quantification of GFP intensity for experiments shown in (K–N, P–S). (U,V) PRDM13 overexpression inhibits *e1Olig::GFP* and *e3Olig::GFP* expression while neither PTF1A or ASCL1 alter levels. Student's t-test was used for significance relative to control except where indicated, **p<0.01, ***p<0.001, ns = not significant. Error bars indicate SEM (n ≥ 8 embryos). (W) Model for how expression of the *Olig* locus is inhibited by PRDM13 in the dorsal neural tube. Loss of this inhibition in the *Prdm13* mutant leads to PTF1A driven ectopic expression. Scale bar: 50 μM.

DOI: https://doi.org/10.7554/eLife.25787.012

The following source data and figure supplement are available for figure 6:

**Source data 1.** PRDM13 is required to block *Olig2* expression in the dorsal neural tube via inhibition of PTF1A.

DOI: https://doi.org/10.7554/eLife.25787.014

**Figure supplement 1.** (A) Genome tracks showing ChIP-Seq data for PTF1A and the 3 PRDM13 experiments at the *Olig2/Olig1* locus.

DOI: https://doi.org/10.7554/eLife.25787.013

the conclusions with regulation of the *Olig* genes, these findings demonstrate a role for PRDM13 in restricting *Prdm12* to the ventral neural tube.

To determine if PRDM13 and PTF1A are directly regulating *Prdm12*, we analyzed the *Prdm12* genomic region for sites bound by these factors. PRDM13 ChIP-seq identified a site, *e1Prdm12*, overlapping a binding site for PTF1A (**Figure 7E** and **Figure 7—figure supplement 1**). *e1Prdm12* contains an E-box, but no compound E-box/TC-box for the PTF1 trimer complex binding (**Figure 7F**). *e1Prdm12::GFP* directs reporter expression localized to the endogenous *Prdm12* domain and is repressed by PRDM13 (**Figure 7G,H,J**). However, co-electroporation with *Ptf1a* had no effect (**Figure 7I,J**). This latter result is consistent with the lack of PTF1 trimer binding motif, which is the motif for the PTF1A activator complex (**Hori et al., 2008**; **Meredith et al., 2009**).

In contrast to the dI4 restricted ectopic expression seen with OLIG2, the ectopic expression of *Prdm12* in the dorsal neural tube of the *Prdm13* mutants was in a pattern reminiscent of two Ebox binding bHLH factors, NEUROG1 and NEUROG2, that are normally present in progenitors to the dI2 and dI6 domains as well as in the ventral neural tube (**Gowan et al., 2001**) (**Figure 7K,M**). Support for these bHLH factors being involved in the ectopic dorsal expression of *Prdm12* was suggested by the RNA-seq data that showed both *Neurog1* and *Neurog2* as being increased in *Prdm13*^GFP/GFP^ mutants (**Table 1**, 3.4- and 1.7-fold respectively). Indeed, mRNA in situ hybridization shows both of these factors increased in the dorsal neural tube of the *Prdm13* mutants (**Figure 7L, N**). We co-electroporated expression constructs for *Neurog1* and *Neurog2* with the *Prdm12* enhancer reporter *e1Prdm12::GFP* and found that both drive activity of the reporter, activity that is suppressed when *Prdm13* is added (**Figure 7O–S**). And finally, we show that PRDM13 interacts with NEUROG1 and NEUROG2 using co-IP assays with lysates from HEK293 cells transfected with these factors (**Figure 7T**).

These data demonstrate the requirement for PRDM13 to restrict *Prdm12* expression to the ventral neural tube (**Figure 7U**). Transcriptional activators such as NEUROG1 and NEUROG2 can activate transcription through an enhancer located more than 25 kb upstream of *Prdm12*. In the absence of PRDM13, there is ectopic expression of *Prdm12* in the dorsal neural tube, expression that may be a consequence of increased NEUROG1 and NEUROG2 combined with the loss of PRDM13.

## Discussion

The ability to generate precise cell identities within an organ is a fundamental process in developmental biology. Central to this process in the developing nervous system are mechanisms that include combinatorial expression of TFs, and TF networks that rely on extensive feedforward and feedback mechanisms to direct cell-type specific gene transcription. To specify the neuronal

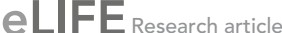

**Figure 7.** PRDM13 is required to restrict *Prdm12* from expression in the dorsal neural tube via inhibition of NEUROG1/2. (A–D) In situ hybridization shows ectopic *Prdm12* mRNA in the dorsal neural tube in *Prdm13* mutants but not *Ptf1a* mutants at E11.5. (E) Genome tracks showing ChIP-Seq data from mouse neural tube tissue for PRDM13 and PTF1A at the *Prdm12* locus. The PRDM13 peaks are highlighted and a dot above the peak indicates if the site was called as significant (see Materials and methods). Region *e1* was tested in reporter assays. (F) The sequence at the center of the PRDM13

*Figure 7 continued on next page*

*Figure 7 continued*

ChIP peak shows *e1* has an Ebox. (G-I, O-R,) Representative images from transverse sections of chick neural tubes co-electroporated with *e1Prdm12::*
*GFP* reporter constructs and expression vectors as indicated at HH13-14 and harvested 24 hr later. Antibodies to MYC-epitope tag or PRDM13 (white)
confirm the electroporation efficiency and the overexpression of the respective transcription factor. (G) In situ hybridization shows endogenous
*cPrdm12*. The dotted lines indicate the V1 domain. (G–I) PRDM13 overexpression inhibits *e1Prdm12::GFP* expression whereas PTF1A has no effect. (O–
R) NEUROG1 and NEUROG2 induce *e1Prdm12::GFP* expression and this is inhibited by PRDM13. (J, S,) Quantification of GFP intensity from chick
electroporation experiments. Student's t-test was used for significance relative to control except where indicated, **p<0.01, ***p<0.001, ns = not
significant. Error bars indicate SEM (n $\geq$ 8 embryos). (K–N) In situ hybridization shows ectopic *Neurog1* and *Neurog2* mRNA in the dorsal neural tube in
the *Prdm13* mutants at E10.5 relative to wildtype. (T) Western blot showing Co-IP of NEUROG1 and NEUROG2 by PRDM13 from HEK293 cells
ectopically expressing these factors. (U) Model for how *Prdm12* expression is inhibited by PRDM13 in the dorsal neural tube. Loss of this inhibition in
the *Prdm13* mutant leads to NEUROG driven ectopic expression of *Prdm12*. Scale bar: 50 µM.
DOI: https://doi.org/10.7554/eLife.25787.015

The following source data and figure supplement are available for figure 7:

**Source data 1.** PRDM13 is required to restrict *Prdm12* from expression in the dorsal neural tube via inhibition of NEUROG1/2.
DOI: https://doi.org/10.7554/eLife.25787.017
**Figure supplement 1.** Genome tracks showing ChIP-Seq data for PTF1A and the 3 PRDM13 experiments at the *Prdm12* locus.
DOI: https://doi.org/10.7554/eLife.25787.016

subtypes in the dorsal spinal cord involved in somatosensory information processing, the transcriptional repressor PRDM13 plays a critical role. We demonstrate its importance in restricting expression of inappropriate regulatory TFs within the lineage that generates the inhibitory neurons of the dorsal spinal cord (summary *Figure 8*). Within the TF network, PRDM13 is a component of a negative feedback loop where the PTF1 trimeric complex activates transcription of *Prdm13* then PRDM13 feeds back to repress transcription of *Ptf1a*, ensuring PTF1A is present only transiently during neural development. Furthermore, a primary mechanism for PRDM13 repression appears to be recruitment to chromatin sites bound by neural bHLH transcription activators, thus providing a molecular explanation for why localization of the bHLH activator does not always result in active gene transcription. Finally, this study demonstrates PRDM13 is required to restrict expression of multiple neuronal specification factors to the ventral neural tube to ensure precision in cell-fate identity in the dorsal spinal cord (*Figure 8* and *Figure 8—figure supplement 1*).

## The PRDM13 ZF domain is required for neural tube neuronal specification and survival

PRDM13, like other PRDM family members, has a PR domain and multiple ZFs (*Hohenauer and Moore, 2012*). PR domains have homology to SET domains that have histone methyltransferase activity, and the ZFs can provide protein-protein and protein-DNA interactions. PRDM13 was shown to contain, or recruit, histone methyltransferase activity (*Hanotel et al., 2014*). Nevertheless, when tested in overexpression assays, truncated versions of the protein lacking the PR domain but containing the terminal three ZFs have similar activity as the full-length protein (*Chang et al., 2013*; *Watanabe et al., 2015*). In addition, a *Prdm13* mutant mouse with the PR domain encoding exons 2 and 3 deleted (referred to here as *Prdm13*$^{\Delta2/3}$) survived to adulthood and exhibited a retinal phenotype (*Watanabe et al., 2015*). Although the neural tube was not examined in these mutants, they demonstrate the PR domain at least is not required for survival.

The *Prdm13* mutants generated here, particularly the *Prdm13*$^{\Delta ZF}$ mutants, highlight the requirement for the C-terminal ZF domain for PRDM13 function in neuronal specification in the neural tube and survival of the animals. While both *Prdm13*$^{GFP}$ and *Prdm13*$^{\Delta ZF}$ lack PRDM13 expression and thus appear to be functional nulls, the *Prdm13*$^{\Delta115}$ mutant produces homozygous offspring that live to adulthood with no notable neural tube defects. The *Prdm13*$^{\Delta115}$ mutant with a 115 bp frameshift deletion upstream of the region encoding the PR domain was predicted to produce a non-functional truncated protein. Surprisingly, immunofluorescence and western analysis using an antibody targeting the C-terminus of the protein showed the presence of a mutant PRDM13 protein that was present at higher levels than the wildtype PRDM13. This suggests a variant PRDM13 is produced, possibly through an aberrant alternative translation start or splicing mechanism, that contains the ZF domain and has sufficient function for survival of the organism. Supporting this premise, the ZF domain has similar activity to the full-length protein in overexpression experiments in the chick



**Figure 8.** Summary of the transcriptional repressor functions of PRDM13 required for correct specification of spinal cord neurons. Three modes of PRDM13 repression that involve different neural bHLH factors (PTF1A, ASCL1, and NEUROG1/2) are diagrammed, as are the wildtype versus the *Prdm13* mutant phenotypes in hemi segments of the E10.5/E11.5 neural tube. (**A**) PRDM13 represses transcription of its activator PTF1A in a negative feedback loop. (**B**) PRDM13 is central to the bistable fate switch between the PAX2+ dI4 inhibitory neurons and the TLX1/3+ dI5 excitatory neurons by repressing transcription of TLX3 via an ASCL1 regulated enhancer. (**C**) PRDM13 functions to exclude ventral neural tube specification factors from being transcribed in the dorsal neural tube.

DOI: https://doi.org/10.7554/eLife.25787.018

The following figure supplement is available for figure 8:

**Figure supplement 1.** Immunohistochemistry for BHLHB5 and in situ hybridization for *Olig1*, *Otp*, *Ccbe1*, *Mfng*, *Phox2a* and *Phox2b* mRNA show ectopic dorsal neural tube expression in the *Prdm13^{ΔZF/ΔZF}* mutants at E11.5 relative to controls.

DOI: https://doi.org/10.7554/eLife.25787.019

neural tube and mouse retina (*Chang et al., 2013*; *Watanabe et al., 2015*). Although *Prdm13^{Δ2/3}* mice have a defect in specification of retinal amacrine interneurons, they survive to adulthood, similar to the *Prdm13^{Δ115}* mutant. We postulate that the *Prdm13^{Δ2/3}* and *Prdm13^{Δ115}* are hypomorphic alleles, and that the ZF domain encoded within exon four is produced in tissues that support survival. Further structural analysis of proteins made in these mutants in a specific tissue manner is required to dissect out the functions of the ZF and PR domains of PRDM13.

## A regulatory feedback loop between PRDM13 and its transcriptional activator PTF1A

PTF1A, essential for specification inhibitory neurons in various regions of the nervous system, directly induces transcription of *Prdm13* in mice, chicken, and Xenopus (*Chang et al., 2013*; *Hanotel et al., 2014*; *Meredith et al., 2013*). PRDM13 in turn feedback inhibits its own expression as reported in Xenopus (*Hanotel et al., 2014*) and detected here in the mouse mutants (*Table 1*), and that of its

activator *Ptf1a*. We previously identified an autoregulatory enhancer for *Ptf1a* located 13–14 kb 5′ of the transcriptional start site (*Masui et al., 2007*; *Masui et al., 2008*; *Meredith et al., 2009*). This enhancer can direct expression of a GFP reporter gene in the correct domain in transgenic mice and in the chick neural tube. Here we show that PRDM13 binds the enhancer in vivo, co-immunoprecipitates with PTF1A, and can block the ability of PTF1A to activate the enhancer in a reporter assay. PTF1A functions as a transcriptional activator in a trimeric complex with an E-protein such as E47 or HEB and RBPJ (*Beres et al., 2006*; *Hori et al., 2008*). PRDM13 may disrupt the complex by displacing RBPJ, however, the precise mechanism has not been determined. The inhibition of *Ptf1a* by PRDM13 represents an incoherent feedback loop, a mechanism present in multiple TF networks (*Alon, 2007*). This mode of feedback inhibition modulates the auto-activation of *Ptf1a* that may be required for its transient expression in the developing nervous system.

## PRDM13 is a repressor of neural bHLH factor activity

Neural bHLH factors such as ASCL1, PTF1A, and NEUROG1/2 bind specific DNA motifs in the genome to regulate gene expression. These factors are transcriptional activators but they can be located near genes that are not transcribed in a given cell. For example, PTF1A is localized to sites around the *Olig1/2* genes but these genes are not transcribed in the dorsal neural tube in the PTF1A domains. Why some genes are expressed and others are not in response to binding of the neural bHLH factors to regulatory regions near these genes was not clear. The neural bHLH factors studied here are referred to a class II bHLH factors as they are cell-type specific and form heterodimers with class I bHLH factors (E-proteins) for function (*Murre et al., 1989*). Class II bHLH activity can be inhibited by ID factors (Inhibitors of DNA binding) that have an HLH domain and can compete with the class II bHLH factors in binding to E-proteins but, without the basic region they do not bind DNA (*Benezra et al., 1990*). PRDM13 represents another mechanism for suppressing activity of these class II bHLH factors.

We propose a model whereby the neural bHLH factors bind target enhancers through E-box motifs, and if PRDM13 is at high enough levels in the cell, it can be recruited to these sites to silence transcription. Multiple lines of evidence support this model including results reported here for PTF1A and NEUROG1/2, and in *Chang et al. (2013)* for ASCL1. (1) PRDM13 and neural bHLH factors like ASCL1 and PTF1A are localized to the same sites in vivo as seen by ChIP-Seq from neural tubes. (2) PRDM13 and multiple class II bHLH factors, but not class I bHLH factors, co-immunoprecipitate. (3) Neural bHLH factors activate transcription through the identified regulatory regions in reporter assays while PRDM13 blocks this activity. (4) Loss of PRDM13 function in vivo results in ectopic expression of multiple bHLH regulated genes in the dorsal neural tube. (5) The primary binding motif identified by PRDM13 ChIP-Seq is an E-box, the motif for bHLH factors. While it is unclear how PRDM13 interacts with bHLH factors and the precise mechanism for specific repression, the findings presented here combined with previous reports provide strong support for PRDM13 interacting with and orchestrating repression of class II bHLH transcriptional activity. Support for the relevance of this mechanism beyond PRDM13 and the bHLH factors studied here has been shown for PRDM8, a PRDM factor closely related to PRDM13, and BHLHB5 in regulating gene expression for proper axon targeting (*Ross et al., 2012*).

Ross *et al*. propose PRDM8 and BHLHB5 form a transcription repressor complex where BHLHB5 is the specific DNA binding factor that recruits PRDM8. In support of this model, PRDM8 does not bind DNA targets in the absence of BHLHB5 (*Ross et al., 2012*). Although not definitively demonstrated, we modeled PRDM13 similarly with it being recruited by bHLH transcription activators such as PTF1A, NEUROG1, NEUROG2, and ASCL1 (*Figures 5–7*, and [*Chang et al., 2013*]). Support for this comes from the primary motif in the PRDM13 bound regions being a bHLH binding motif (*Figure 4*). Nevertheless, the ZF domain is predicted to have DNA binding properties suggesting additional experiments testing this aspect of the model are required to determine if PRDM13 directly interacts with DNA in a site-specific manner or not.

How PRDM13 actually represses the activity of the neural bHLH factors is not clear. Some PRDM proteins function through intrinsic histone methyltransferase (HMT) activity dependent on the PR domain of the protein (*Eom et al., 2009*; *Eram et al., 2014*; *Huang et al., 1998*; *Pinheiro et al., 2012*; *Wu et al., 2013*; *Wu et al., 2008*). Those PRDM members that lack intrinsic HMT activity depend on the recruitment of histone modifying co-factors that are necessary for transcriptional repression (*Bikoff et al., 2009*; *Chittka et al., 2012*; *Ross et al., 2012*). PRDM13 has been

demonstrated to have histone methyltransferase activity (*Hanotel et al., 2014*), however, whether this activity is intrinsic to the protein or is through recruitment of a methyltransferase is not clear. The mutant alleles discussed above suggest the PR domain may not be required for activity, at least for cell-type specification in the neural tube and processes required for survival. While our study provides a molecular pathway for repression directed by PRDM13, further study is needed to identify the mechanism of repression for this PRDM factor.

### Dorsal spinal cord neurons require PRDM13 for repression of specification factors for ventral identity

Decades of work have gone into defining progenitor domains and resulting neuronal populations in the developing spinal cord mainly by combinatorial TF expression of HD and bHLH factors (*Lai et al., 2016*). Repress of inappropriate gene expression programs in a lineage is critical to specify appropriate cell fate and provide precision in cell-type identify. Cross-repression between TFs is a major principle in setting up boundaries that delineate either progenitor domains or their resulting neurons. This premise was first observed in the ventral neural tube where neighboring progenitors were found to use HD factors to repress each other's' expression to generate discrete progenitor boundaries (*Briscoe et al., 2000*; *Ericson et al., 1997*). In the dorsal neural tube, cross-repression is also evident and has been shown to occur between bHLH factors (*Glasgow et al., 2005*; *Gowan et al., 2001*; *Mizuguchi et al., 2006*; *Wildner et al., 2006*). Until recently, the cross-repressive mechanisms elucidated in the developing neural tube have been limited to gene programs in neighboring progenitor populations. However, two recent studies of the ventral neural tube TFs NKX2.2, NKX6.1 and OLIG2 uncover broader programs of repression than previously appreciated and demonstrate that these transcription factors directly repress all alternative fates, including dorsal cell fate programs, in the ventral neural tube (*Kutejova et al., 2016*; *Nishi et al., 2015*). Thus, progenitor-specific transcriptional repressors limit the activity of broadly expressed transcriptional activators to generate progenitor diversity. This principle of cell-type specific repressors overcoming broader activation of genes for precision in cell identity is used in regulating expression of terminal differentiation genes as well (*Kerk et al., 2017*).

Here we provide evidence for an additional variation of these models whereby the transcriptional repressor is broadly expressed and it limits inappropriate gene expression driven by progenitor-specific activators, the bHLH factors. PRDM13 is identified as a key transcriptional repressor of a whole battery of genes, not just those in domains adjacent to the domain giving rise to the dorsal inhibitory neurons, but also factors normally restricted to the ventral neural tube and required for neuronal identity in that region (*Table 1*) (*Figure 8* and *Figure 8—figure supplement 1*). PRDM13 repressor activity is required to limit the bHLH driven transcription of these ventral specification factors. In particular, we highlight OLIG2 and PRDM12, which are normally restricted to the ventral neural tube but in the *Prdm13* mutants they are aberrantly present dorsally, altering the identity of these neurons. Taken together, PRDM13 is a transcriptional repressor essential in the dorsal spinal cord development to modulate the transcriptional activity of neural bHLH factors to generate precise neuronal identities.

## Materials and methods

Further information and requests for resources and reagents should be directed to and will be fulfilled by the Lead Contact Jane E. Johnson (jane.johnson@utsouthwestern.edu).

### Mouse strains

The *Prdm13^GFP^* mutant and the *Prdm13^fusionGFP^* mouse model were developed using zinc-finger nuclease technology (*Figure 1D*). mRNA encoding two zinc-finger nuclease (ZFN) proteins targeting within the first exon of the *Prdm13* gene (CACCAGCGTGAACGCTGActgctGCATCCCGGCTGGCT) were purchased from Sigma-Aldrich (St. Louis, MO, USA) and delivered by pronuclear injection into fertilized mouse eggs along with a donor plasmid. For *Prdm13^GFP^* mutant mice, the donor plasmid encoded GFP followed by a stop codon, and contained two homology regions each 750 bp in length to allow for homologous recombination of the insert into the *Prdm13* locus (*Figure 1D*). Out of 35 potential founders, eight integrated the GFP coding region at the designed site as verified by sequencing the locus. Two independently generated strains were bred. Initial phenotypes were

identical including 100% penetrance of P0 lethality and a loss of PAX2 in the dorsal neural tube at E10.5 in homozygous mutant mice. Subsequently, these strains were interbred for all experiments shown here. A similar donor plasmid without the stop codon was used to generate the *Prdm13*[fusionGFP] mouse that was used for ChIP-seq. Homozygous *Prdm13*[fusionGFP] mice live to adulthood and show no changes in PAX2 or TLX1/3 expression at E10.5 (data not shown). Mice were genotyped by PCR with 5'-GCTGCTCCTGGTTCTGTCA-3', 5'-CCTTTTCTCTGCTGCTCGTC-3' and 5'-GCTGGAG TACAACTACAACAGCCA-3' (wild-type 313 bp; mutant 549 bp).

The *Prdm13*[Δ115] mutant line is a 115 bp deletion generated within the first exon of *Prdm13* upon injection of the ZFN mRNA (chr4:21,612,669–21,612,783, mm10). Homozygous *Prdm13*[Δ115/ Δ115] mice live to adulthood, and show no changes in PAX2 or TLX1/3 expression at E10.5. PCR genotyping used primers 5'-GCTGCTCCTGGTTCTGTCA-3' and 5'-CCTTTTCTCTGCTGCTCGTC-3' (wild-type 313 bp; mutant 198 bp) (*Figure 1E*).

The *Prdm13*[ΔZF] mutant mouse line was generated by pronuclear injection of CAS9 mRNA and two sgRNA targeting the 3' end of *Prdm13* (*Figure 1D*). Out of 39 potential founders we obtained one line with a 454 bp deletion in the fourth exon of *Prdm13* that results in a protein truncated at amino acid 602 and deletes the three C-terminal zinc-fingers of PRDM13, adding 25 nonsense amino acids in the terminal portion of the protein. Once bred to homozygosity these mice die at P0 and phenocopy all changes observed in the *Prdm13*[GFP/GFP] mutants at all developmental stages evaluated. Mice were PCR genotyped with 5'-GATCGCCATGCACACACAGC-3' and 5'-CAATGAAGCCC TTCTTGT-3' (wild-type 617 bp; mutant 207 bp). The *Ptf1a*[CRE] mouse line replaces the coding sequence for *Ptf1a* with that for *Cre* recombinase (*Kawaguchi et al., 2002*). Genotyping was performed as previously described (*Glasgow et al., 2005*).

All animal work was approved by the Institutional Animal Care and Use Committee at UT Southwestern. All mouse strains are maintained on a mixed background of ICR and C57Bl/6.

## Tissue preparation, immunohistochemistry and in situ hybridization

Mouse embryos were dissected in ice-cold 0.1M sodium phosphate buffer pH 7.4 and fixed in 2% paraformaldehyde for 1 hr at 4° C. For E16.5, spinal cords were dissected out of the embryo prior to fixation. Tissue was washed 3X in ice-cold 0.1M sodium phosphate buffer pH 7.4 for 15 min and sunk overnight in 30% sucrose in PBS for E10.5 and E11.5 embryos, 15% sucrose in PBS for E12.5 embryos and 30% sucrose in water for E16.5 embryos. Tissue was embedded in Tissue-Plus O.C.T. compound (Fisher Healthcare, Houston, TX, USA) and cryosectioned at 20 μm (E10.5-E11.5) and 30 μm (E12.5-E16.5) focusing on sections from the upper limb level.

For in ovo chick electroporation assays, fertilized White Leghorn eggs were obtained from the Texas A and M Poultry Department (College Station, TX, USA) and incubated for 48 hr at $37^0$C. The supercoiled reporter plasmids were diluted to 1.5 mg/ml in $H_20$/1X loading dye and injected into the lumen of the closed neural tube at stages HH13-15 along with either a pMiWIII-Myc epitope tagged vector serving as an electroporation control or a TF containing expression vector. The injected embryos were then electroporated with 5 pulses of 25 mV each for 50 msec with intervals of 100 msec. Embryos were harvested 48 hr later at stages HH22-23, fixed with 4% paraformaldehyde for 45 min, and processed for cryosectioning and immunofluorescence.

Immunofluorescence was performed using the following antibodies: guinea pig anti-PRDM13 (1:1000, [*Watanabe et al., 2015*]), mouse anti-LHX1/5 (1:100, DSHB Hybridoma Product 4F2 deposited to the DSHB by Jessell, T.M./Brenner Morton, S.), rabbit anti-PAX2 (1:500, Invitrogen, Carlsbad, CA, USA, #71–6000), guinea pig anti-TLX1/3 (1:10000, [*Müller et al., 2002*]), guinea pig anti-PTF1A (1:10000, TX507 [*Hori et al., 2008*]), guinea pig anti OLIG3 (*Müller et al., 2005*), guinea pig anti BHLHB5 (*Ross et al., 2012*), rabbit anti-OLIG2 (1:1000, Millipore, Billerica, MA, USA, #AB9610), rabbit anti-OLIG2 (1:1000, [*Wang et al., 2006*]) and guinea pig anti-ASCL1 (1:10000, TX518 [*Kim et al., 2008*]). Imaging was performed with a ZEISS LSM 510 confocal microscope.

In situ hybridization was performed as previously described (*Meredith et al., 2009*). Probes used were mouse *Prdm13* and *Ptf1a* (*Chang et al., 2013*), mouse *Prdm12* (*Thélie et al., 2015*), mouse *Gad1* and *Vglut2 (Slc17a6)* (*Cheng et al., 2004*), mouse *Otp*, *Phox2a* and *Phox2b* (*Gray et al., 2004*), mouse *Olig1* (*Zhou and Anderson, 2002*), chick *Neurog1* and *Neurog2* probes (gift from Jessell lab). The chick *Prdm12* probe was generated starting from a pBSK+ *cPrdm12* cDNA (EST BU233582). Probes for *Ccbe1* and *Mfng* were generated by PCR from mouse neural tube cDNA. Imaging was performed with a Hamamatsu Nanozoomer 2.0HT digital slide scanner.

## Generation of reporter constructs and expression vectors

Regulatory elements bound by PRDM13 were cloned into the MCSIII GFP reporter cassette. This reporter cassette contains the β-globin minimal promoter, a nuclear localized fluorescence reporter, and the 3' cassette from the human growth hormone. Mammalian conservation from the UCSC genome browser was used to identify boundaries of the elements cloned. All regions were PCR amplified from ICR mouse DNA. Genomic coordinates for regions cloned are based on the mouse mm10 genome build and include *e1Prdm12* (chr2:31,609,785–31,610,285), *e2Prdm12* (chr2:31,640,634–31,641,503), *e1Olig* (chr16:91,140,306–91,140,968), *e2Olig* (chr16:91,246,035–91,247,122), *e3Olig* (chr16:91,333,568–91,335,428). *E1Olig-EboxMut, e2Olig-PTF1Mut* and *e2Olig-allEboxMut* were subcloned using gBlocks with E-box mutations (Integrated DNA Technologies, Coralville, IA, USA). KpnI/SpeI restriction sites were introduced on each end for cloning adjacent to the β-globin minimal promoter in the reporter cassette. The *Ptf1a* 2.3 kb enhancer reporters are as described (*Meredith et al., 2009*). TFs PRDM13, $^{myc}$PTF1A, $^{myc}$ASCL1, $^{myc}$NEUROG1, and NEUROG2 were expressed in the pMiWIII expression vector (*Chang et al., 2013*; *Gowan et al., 2001*; *Matsunaga et al., 2001*). All constructs were sequence verified and expression of the TFs confirmed by immunohistochemistry with antibodies to the myc tag or with TF-specific antibodies.

## Co-immunoprecipitation assays and western blotting

HEK293 cells were transfected using FuGENE six reagent (Promega, Madison, WI, USA) with pMiWIII (*Matsunaga et al., 2001*) expression vectors for $^{myc}$HOOK3, $^{myc}$PTF1A, $^{myc}$ASCL1, $^{myc}$NEUROG1, $^{myc}$E47, $^{myc}$E12, $^{myc}$HEB, and NEUROG2, or PRDM13$^{FLAG}$ in pCIG (*Chang et al., 2013*; *Gowan et al., 2001*; *Hori et al., 2008*), and total cell lysate was prepared with an IP lysis buffer (Thermo Fisher Scientific, Waltham, MA, USA) after 48 hr of incubation. A/G beads conjugated with anti-FLAG antibodies (Sigma-Aldrich, #A2220) were used for immunoprecipitation, and mouse anti-cMYC (1:1000, Santa Cruz Biotechnology, Dallas, TX, USA, #sc-789), mouse anti-FLAG M2 (1:1000, Sigma, #F3165), rabbit anti-NEUROG1 (1:500, T2925, [*Gowan et al., 2001*]), or rabbit anti-NEUROG2 (1:2000, T2923, Johnson lab generated, antigen was domain N-terminal to the bHLH domain of mouse NEUROG2) were used to probe the resulting Western blots. Some blots were stripped and reprobed. Western lightning plus-ECL (Perkin Elmer, Waltham, MA, USA) was used for luminescence signal detection. For detection of endogenous PRDM13, E11.5 mouse neural tubes or telencephalon tissue were collected from wild-type or *Prdm13* mutant embryos, protein lysates prepared with RIPA lysis buffer, and Western blotted with anti-PRDM13 antibodies recognizing the C-terminal (685–754 amino acids) (*Watanabe et al., 2015*). The HEK293 cells, originally from ATCC, have not recently been authenticated or tested for mycoplasma. These cells were used for co-immunoprecipitation of ectopic expression of epitope tagged proteins, internally controlled experiments not impacted by possible cell line misidentification.

## mRNA isolation and RNA-Sequencing

Mouse neural tubes dissected from E11.5 *Prdm13$^{GFP}$* embryos either heterozygous or homozygous for GFP were individually placed into cold DMEM/F12 and dissociated in 0.25% trypsin for 15 min at 37 degrees C. Trypsin activity was quenched with 2% fetal bovine serum, and GFP positive cells were purified from the resulting single cell suspension by fluorescence activated cell sorting (FACS). Total RNA from FACS isolated cells was extracted and purified with Mini RNA Isolation Kit (Zymo Research, Irvine, CA, USA). Genotypes were determined and RNA from 9 homozygous and 7 heterozygous embryos were combined, mRNA was purified, reverse transcribed and amplified for sequencing with Illumina's mRNA-Seq kit (Illumina, San Diego, CA, USA. Two independent libraries representing technical replicates were sequenced for each cell population.

Sequence reads from RNA-seq were assembled using mm10 refSeq gene annotation with Tophat2 (v.2.1.0) (*Kim et al., 2013*). Gene expression abundance was estimated by counting the number of reads that mapped to any exon of a given gene using featureCounts (v.1.5.0-p1) from subread package (*Liao et al., 2014*). Counted reads were normalized using the TMM method (*Robinson and Oshlack, 2010*) and differential expression analysis was performed using the edgeR package (v.3.6.8) (*Robinson et al., 2010*). For a gene to be called as differentially expressed, it required a p-value<0.05 and had to have an FPKM >1 in either the mutant or control sample.

## Chromatin immunoprecipitation, sequencing and qPCR

PTF1A and ASCL1 ChIP-Seq were previously published (*Castro et al., 2011*; *Meredith et al., 2013*) (GSE55840). For PRDM13 ChIP-seq using PRDM13-Ab1 (ZF) and PRDM13-Ab2 (FL), neural tubes from multiple litters of E11.5 ICR embryos were collected, as were telencephalon from the same embryos as a negative control. E11.5 *Prdm13^fusionGFP* embryos were used for GFP ChIP. ChIP was performed as previously described (*Meredith et al., 2013*). Notably, the PRDM13 ChIP – Ab2 (FL) was treated with formaldehyde along with the long-arm cross-linker, ethylene glycol-bis (succinimidyl succinate), to increase ChIP efficiency (*Zeng et al., 2006*). Libraries were made according to Illumina's ChIP-Seq DNA sample preparation protocol. Rabbit antibodies against PRDM13 were Johnson lab generated (PRDM13-Ab1 (ZF) PA6659, rabbit anti-PRDM13 and PRDM13-Ab2 (FL) TX970, rabbit anti-PRDM13). The antigens were bacterially expressed C-terminal domain of PRDM13 including amino acids 622 to 755 or full length protein, respectively.

Sequence reads for each sample were mapped to the mm10 genome assembly by using Bowtie2 (v.2.2.6) (*Langmead and Salzberg, 2012*). Mapped reads were filtered to remove low quality reads using samtools (v.1.2) (*Li et al., 2009*). Duplicate reads were removed using picard tools (v.1.119), and the remaining unique reads were normalized to 10 million reads. Peak calling was performed by HOMER (v.4.7) using an FDR cutoff of 0.001, a cumulative Poisson p-value of <0.0001, and required a 4-fold enrichment of normalized sequence reads in the treatment sample over the control/input sample (3-fold was used for the GFP ChIP-seq). To determine PRDM13 specific peaks, we compared the PRDM13 bound sites identified in PRDM13 ChIP samples using three different antibodies. A common binding site between two samples was called when the peak summits of each sample were found within 150 bp of each other. We used PRDM13 bound sites that are shared in two out of three samples for all downstream analysis. We used a 150 bp distance cutoff for comparison of PRDM13 peaks with ASCL1 and PTF1A peaks. Distance to gene and gene annotations for ChIP-Seq peaks were obtained using GREAT v3.0 (*McLean et al., 2010*). GREAT assigns a gene to a binding region if the region falls within 5 kb 5' or 1 kb 3' of the transcription start site (basal region), with a maximum extension of 1000 kb in either direction. If the binding region falls within the basal region of multiple genes, then more than one assignment is made. All parameters were left at their default settings. De novo motif analysis was performed in 150 bp around the peak summit using findMotifsGenome module in HOMER (v.4.7). GO analysis was performed using ConsensusPathDB (*Kamburov et al., 2013*).

## Quantification and statistical analysis

In sections from mouse embryos with cells labeled by immunofluorescence (*Figures 2* and *3*), quantification of cell number was assisted by ImageJ software on five or more embryos for each genotype (n = number of embryos, n $\geq$ 5, biological replicates). For analysis of GFP intensity in chick electroporation experiments (*Figures 5–7*), sections from at least eight different embryos (n = number of embryos, n $\geq$ 8, technical replicates) were analyzed and used for quantification. Only sections with confirmed electroporation across the dorsal ventral axis were used in the analysis. Mean GFP pixel intensity for *enh::GFP* expression was measured by ImageJ on the electroporated side of the neural tube with the background levels measured from the non-electroporated side subtracted. Significant differences between control and experimental samples in each case were calculated using a two-tailed two-sample unequal variance (homoscedastic) Student's t-test in Microsoft Excel. *p<0.05, **p<0.01, ***p<0.001, ns = not significant. Error bars indicate SEM. The enrichment of PRDM13 ChIP targets in the list of genes that showed significant expression change in mutant samples relative to controls was calculated using the hypergeometric test.

## Data and software availability

RNA-Seq and ChIP-Seq data generated here are available on the GEO database (GSE90938).

## Acknowledgements

We acknowledge the many hours of helpful discussions with members of the Johnson laboratory, critical reading of the manuscript by Drs. R MacDonald, H Lai, and J Wu, and technical support from E Kibodeaux, J Villarreal, and T Savage. We are grateful for the excellent transgenic mouse services

provided by the UT Southwestern Transgenic Core (Dr. R Hammer, Director), and the generous gifts from Drs. T Furukawa (PRDM13 antibodies), T Muller and C Birchmeier (TLX1/3 and OLIG3 antibodies), QR Lu (OLIG2 antibodies), S Ross (BHLHB5 antibodies) and E Bellefroid (mouse *Prdm12* in situ probe). This work was supported by the National Institutes of Health R01 HD037932, R37 HD091856, and R01 NS032817 to JEJ.

# Additional information

## Funding

| Funder | Author |
| --- | --- |
| National Institutes of Health | Jane E Johnson |

The funders had no role in study design, data collection and interpretation, or the decision to submit the work for publication.

## Author contributions

Bishakha Mona, Investigation, Methodology, Writing—original draft; Ana Uruena, Conceptualization, Investigation, Methodology, Writing—original draft; Rahul K Kollipara, Software, Formal analysis, Writing—review and editing; Zhenzhong Ma, Mark D Borromeo, Investigation, Writing—review and editing; Joshua C Chang, Investigation; Jane E Johnson, Conceptualization, Resources, Supervision, Funding acquisition, Methodology, Writing—original draft

## Author ORCIDs

Jane E Johnson http://orcid.org/0000-0002-8605-2746

## Ethics

Animal experimentation: This study was performed in strict accordance with the recommendations in the Guide for the Care and Use of Laboratory Animals of the National Institutes of Health. All animal work was approved by the Institutional Animal Care and Use Committee at UT Southwestern protocol 2015-101400 and 2015-101338.

## Decision letter and Author response

Decision letter https://doi.org/10.7554/eLife.25787.027
Author response https://doi.org/10.7554/eLife.25787.028

# Additional files

## Supplementary files

• Supplementary file 1. ChIP_seq_ZF sheet: Contains all Prdm13 bound regions in E11.5 mouse (ICR) neural tube from ChIP-Seq using Prdm13 Zinc finger antibody. ChIP_seq_GFP sheet: Contains all Prdm13 bound regions in E11.5 mouse (Prdm13-GFP fusion mouse line) neural tube from ChIP-Seq using GFP antibody.ChIP_seq_FL sheet: Contains all Prdm13 bound regions in E11.5 mouse (ICR) neural tube from ChIP-Seq using full length Prdm13 antibody.ChIP_seq reads were mapped to mm10 genome build. Peaks were called using neural tube input for ZF and GFP samples, and telencephalon tissue for FL samples. Maximum of two target genes were assigned to each peak using GREAT software. The chromosome and the coordinates for the start and end are given for each peak. Data available GSE90938. Peaks-shared sheet: PRDM13 bound peaks called in at least 2 of the three experiments. RNA_seq sheet: Contains normalized expression (FPKM) values obtained from RNA-seq experiments performed on GFP sorted cells from E11.5 mouse neural tube (Prdm13-GFP/GFP = Prdm13 homo and Prdm13-GFP/+ = Prdm13 het). (I) An fpkm value of >1 was used to determine expression in the sample population.
DOI: https://doi.org/10.7554/eLife.25787.020

• Supplementary file 2. RNA-seq from Prdm13 Het samples (Prdm13GFP/+) and Prdm13 Homo samples (Prdm13GFP/GFP). GFP cells were isolated by FACS from E11.5 neural tubes. Up_reg_in_HET_1.5FC sheet contains genes that are upregulated (Fold change >= 1.5; FPKM >= 1) in Prdm13GFP/+compared to Prdm13GFP/GFP. Down_reg_in_HET_1.5FC sheet contains genes that are downregulated (Fold change <= 1.5; FPKM >= 1) in Prdm13GFP/+compared to Prdm13GFP/GFP. Attached sheets contain over represented pathways in differentially expressed genes (DEGs) that are bound by Prdm13 and just DEGs alone.

DOI: https://doi.org/10.7554/eLife.25787.021

• Supplementary file 3. All PRDM13 ChIP peak-to-gene sheet contains the list of genes associated with PRDM13 binding sites and their expression data from the RNA-seq data inE11.5 neural tubes GFP sorted cells from Prdm13 wt vs mutants (from *Table 1*). For these targets, PRDM13 binding site information was provided and also whether ASCL1 or PTF1A binding sites were shared with PRDM13 binding sites or not.PRDM13 targets sheet contains only those genes from the previous list that were differentially expressed in the Prdm13 wt vs mutants. The genes normally repressed by PRDM13 are highlighted in red and the genes induced by PRDM13 are highlighted in green.

DOI: https://doi.org/10.7554/eLife.25787.022

## Major datasets

The following dataset was generated:

| Author(s) | Year | Dataset title | Dataset URL | Database, license, and accessibility information |
|---|---|---|---|---|
| Mona B, Uruena A, Kollipara RK, Ma Z, Borromeo MD | 2016 | Repression by PRDM13 is critical for generating precise neuronal identity | https://www.ncbi.nlm.nih.gov/geo/query/acc.cgi?acc=GSE90938 | Publicly available at the NCBI Gene Expression Omnibus (accession no: GSE90938) |

The following previously published dataset was used:

| Author(s) | Year | Dataset title | Dataset URL | Database, license, and accessibility information |
|---|---|---|---|---|
| Borromeo MD, Meredith DM, Castro DS, Chang JC, Tung K, Guillemot F, Johnson JE | 2014 | Transcription Factor Network Specifying Inhibitory versus Excitatory Neurons in the Dorsal Spinal Cord [ChIP-Seq] | https://www.ncbi.nlm.nih.gov/geo/query/acc.cgi?acc=GSE55840 | Publicly available at the NCBI Gene Expression Omnibus (accession no: GSE55840) |

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
