## [Decision Letter]

Thank you for submitting your article "Repression by PRDM13 is critical for generating precision in neuronal identity" for consideration by *eLife*. Your article has been reviewed by three peer reviewers, and the evaluation has been overseen by a Reviewing Editor and Didier Stainier as the Senior Editor. The following individuals involved in review of your submission have agreed to reveal their identity: Diogo S Castro (Reviewer #1); John LR Rubenstein (Reviewer #2).

The reviewers have discussed the reviews with one another and the Reviewing Editor has drafted this decision to help you prepare a revised submission

The described study utilizes newly generated null mutations in Prdm13 to investigate the function of this transcriptional regulator in the developing dorsal spinal cord. Resulting data supports previous findings that Prdm13 suppresses the fate of excitatory neurons in inhibitory lineages. By a combination of developmental and genomic assays, the authors demonstrate that the recruitment of PRDM13 to bHLH bound enhancers inhibits the expression of ventral transcription factors in the dorsal neural tube. While consistent with current models of neural tube patterning this study provides significant mechanistic insight into the Prdm13-mediated gene regulatory network that regulates neurogenesis in the dorsal spinal cord.

While a carefully performed study, interpretation of the data would be further strengthened by additional analysis which must be adequately addressed in a revised manuscript.

1) Prdm13 ChIP-seq peaks do not show enrichment for a specific motif but instead for several sequences recognized by various neural transcription factors, suggesting Prdm13 is indirectly recruited to DNA. Thus, it become even more important to check the specificity of the Prdm13 antibody. This should be done by validating a series of peaks (associated with the various DNA motifis) in chromatin from wild type and Prdm13 null embryos.

2) In Figure 5, the activity of the e2 and e3 enhancer is not restricted to the olig2 domain. To confirm the specificity of the assay, the authors should make a point mutation in the binding motif and show that it abolishes the expression of GFP in the electroporated neural tube. This would demonstrate that the reporter activity is due to the activity of the regulatory region cloned, even if does not reflect the endogenous expression of the putative target gene.

3) Although the authors conclude that "these findings highlight the function of PRDM13 in repressing bHLH transcriptional activators" which implicitly includes Ascl1, the manuscript does not provide an example of a validated Ascl1/Prdm13 target. The authors should rewrite the conclusion accordingly, so that it is not misleading.

---

## [Author Response]

*While a carefully performed study, interpretation of the data would be further strengthened by additional analysis which must be adequately addressed in a revised manuscript.*

We thank the reviewers for their time and suggestions for improving this study. As you will see below, in performing additional analysis in response to the reviewer’s comments we have significantly strengthened our findings and thus, support for our conclusions.

*1) Prdm13 ChIP-seq peaks do not show enrichment for a specific motif but instead for several sequences recognized by various neural transcription factors, suggesting Prdm13 is indirectly recruited to DNA. Thus, it become even more important to check the specificity of the Prdm13 antibody. This should be done by validating a series of peaks (associated with the various DNA motifis) in chromatin from wild type and Prdm13 null embryos.*

We agree it is important to strengthen the data identifying PRDM13 bound sites in the genome and provide some validation for these sites. While we would have liked to provide data from the suggested experiment, we have been unable to obtain efficient enough ChIP enrichment from the limited source material provided by *Prdm13* mutant neural tubes. Instead we offer a different set of experiments that we believe serves a similar purpose and provides a high confident set of PRDM13 target genes. We now identify PRDM13 target genes based on 3 PRDM13 ChIP-seq data sets that were generated with 3 independent strategies. These include: 1) ChIP-seq data from the original submission that is from E11.5 neural tube tissue using a lab produced antibody generated against the terminal ZnF domain of mouse PRDM13. 2) ChIP-seq data from E11.5 neural tube tissue using an independently produced antibody generated against full-length mouse PRDM13. Here, telencephalon, a PRDM13 negative tissue, was used to control for non-specific binding of the antibody. 3) We generated a mutant mouse strain with a GFP fused in-frame N-terminal to the PR domain of PRDM13. Mice homozygous for this GFP-PRDM13 fusion are indistinguishable from wildtype. ChIP-seq data from E11.5 neural tube tissue from these animals using an antibody directed against GFP was generated.

This exercise had a substantial positive impact on the support for the conclusions reported. We restricted our analysis to PRDM13 bound sites that were called as peaks in at least 2 of the 3 ChIP-seq datasets. This limited the number of PRDM13 target genes identified. The genes highlighted in the study including *Ptf1a, Olig2*, and *Prdm12* were all confirmed as PRDM13 targets providing the validation requested by the Reviewers. We show the genome browser tracks for all three ChIP-seq experiments in supplementary data associated with Figure 5–Figure 7, and provide all the called peaks in Supplementary file 1.

This analysis also addresses Reviewer1’s point #2 about PRDM13 as a repressor versus an activator. 54 of 455 genes that increased in the *Prdm13* mutants relative to controls were identified as PRDM13 targets (p-value=1.26e-27), whereas only 6 of 356 genes that decreased in the *Prdm13* mutants were identified as targets (p-value=0.16). These findings support the role of PRDM13 as a repressor (Figure 4, Table 1, Supplementary file 3).

*2) In Figure 5, the activity of the e2 and e3 enhancer is not restricted to the olig2 domain. To confirm the specificity of the assay, the authors should make a point mutation in the binding motif and show that it abolishes the expression of GFP in the electroporated neural tube. This would demonstrate that the reporter activity is due to the activity of the regulatory region cloned, even if does not reflect the endogenous expression of the putative target gene.*

As suggested by the reviewers, we tested the requirement for the E-box sequences in the PRDM13 bound enhancers for *Olig1/2*. We did this for *e1Olig* and *e2Olig* rather than *e3Olig* for reasons described below, and because *e3Olig* was not responsive to activation by PTF1A or ASCL1. These experiments demonstrate that the E-boxes are required for *e1Olig* and *e2Olig* enhancer activity. In the process of performing these experiments we corrected an error in the data reported in the original manuscript. The *e1Olig* sequence originally tested was not the correct sequence (consistent with its lack of activity in the reporter assay). In the current revised manuscript, we corrected this error. None of the 3 *Olig* enhancers direct reporter expression restricted in the endogenous *Olig* domains. Nevertheless, *e1Olig* and *e2Olig* require the Ebox motifs within the enhancers for their activity (Figure 6 and Figure 6—figure supplement 1). Notably, *e2Olig* that has the PTF1 trimer motif requires this motif for enhancer activity and for its responsiveness to PTF1A. The corrected *e1Olig* enhancer behaves similarly to *e3Olig* in that it is repressed by PRDM13 but is not activated by either PTF1A or ASCL1. As the reviewers pointed out, even though the e1 and e2 enhancers are not restricted to the endogenous *Olig* expression domain, the reporters require the E-box motifs for activity.

*3) Although the authors conclude that "these findings highlight the function of PRDM13 in repressing bHLH transcriptional activators" which implicitly includes Ascl1, the manuscript does not provide an example of a validated Ascl1/Prdm13 target. The authors should rewrite the conclusion accordingly, so that it is not misleading.*

We disagree that this statement in the abstract is misleading. In this manuscript, we explicitly provide evidence for PRDM13 repressing multiple bHLH activators including PTF1A, NEUROG1, and NEUROG2. We don’t explicitly show a validated ASCL1/PRDM13 target here, but we have demonstrated this for *Tlx3* previously in Chang et al., 2013 and this is described and referenced where relevant. In the current study, we provide additional support for those conclusions showing the increase in *Tlx3* in the *Prdm13* mutant mice. In the revised manuscript, we now point out that although this current study does not show an example of a validated ASCL1/PRDM13 target, this was shown in Chang et al., 2013 for regulation of TLX3 at the site highlighted in Figure 4. See subsection “PRDM13 is a repressor of neural bHLH factor activity”.